# Research Progress in Conversion of CO_2_ to Valuable Fuels

**DOI:** 10.3390/molecules25163653

**Published:** 2020-08-11

**Authors:** Luyi Xu, Yang Xiu, Fangyuan Liu, Yuwei Liang, Shengjie Wang

**Affiliations:** Centre for Bioengineering and Biotechnology, China University of Petroleum, Qingdao 266580, China; xuluyi1120@163.com (L.X.); xiuyangswhg@163.com (Y.X.); 15069873749@163.com (F.L.); 17854119142@163.com (Y.L.)

**Keywords:** CO_2_ conversion, electrocatalysis, photocatalysis, photoelectrocatalysis, biocatalysis, macrocycles

## Abstract

Rapid growth in the world’s economy depends on a significant increase in energy consumption. As is known, most of the present energy supply comes from coal, oil, and natural gas. The overreliance on fossil energy brings serious environmental problems in addition to the scarcity of energy. One of the most concerning environmental problems is the large contribution to global warming because of the massive discharge of CO_2_ in the burning of fossil fuels. Therefore, many efforts have been made to resolve such issues. Among them, the preparation of valuable fuels or chemicals from greenhouse gas (CO_2_) has attracted great attention because it has made a promising step toward simultaneously resolving the environment and energy problems. This article reviews the current progress in CO_2_ conversion via different strategies, including thermal catalysis, electrocatalysis, photocatalysis, and photoelectrocatalysis. Inspired by natural photosynthesis, light-capturing agents including macrocycles with conjugated structures similar to chlorophyll have attracted increasing attention. Using such macrocycles as photosensitizers, photocatalysis, photoelectrocatalysis, or coupling with enzymatic reactions were conducted to fulfill the conversion of CO_2_ with high efficiency and specificity. Recent progress in enzyme coupled to photocatalysis and enzyme coupled to photoelectrocatalysis were specially reviewed in this review. Additionally, the characteristics, advantages, and disadvantages of different conversion methods were also presented. We wish to provide certain constructive ideas for new investigators and deep insights into the research of CO_2_ conversion.

## 1. Introduction

Overreliance on fossil fuels have led to the discharge of more and more carbon dioxide (CO_2_) into the atmosphere accompanied with the rapid development of modern industry. This has resulted in serious environmental problems, including global warming and other related problems, such as rising sea levels, ocean acidification, ozone layer depletion, and extreme weather patterns [1]. Additionally, the excessive consumption of fossil fuels and deforestation have interrupted the carbon cycle on the earth and accelerated the environmental deterioration and resource depletion [2,3]. In order to mitigate the issue of global warming, most countries have signed the Paris Agreement, which aims to abate the net atmospheric CO_2_ levels by 2050. Therefore, efficient measures have to be made to reduce atmospheric CO_2_ levels, such as reducing the direct emission of CO_2_ by developing clean and sustainable energy resources to replace traditional fossil fuels, capturing and storing CO_2_, and converting the anthropogenic CO_2_ into useful chemicals or fuels by reducing it. Among them, the efficient conversion of CO_2_ has attracted great consideration because of the fact that it is the most abundant C1 compound and is the major greenhouse gas [4,5,6,7]. The conversion of CO_2_ to value-added fuels or chemicals is important in recycling carbon species and simultaneously solving environmental problems and energy crises. 

CO_2_ is one of the most stable molecules in which carbon is in the highest valence state. It is difficult to have an electrophilic reaction because of its poor electron affinity. Hence, the conversion of CO_2_ depends on nucleophilic attack of the carbon atom. As is known, the dissociation energy for breaking the C=O bond in CO_2_ molecules is higher than 750 kJ mol^−1^ [8]. This is an uphill reaction from a thermodynamic point of view. To complete such a reaction, high temperature, high pressure environment, or highly efficient catalysts are typically required to provide the necessary energy. Till now, different strategies including thermal catalysis [9,10,11,12,13], photocatalysis [14,15,16,17], electrocatalysis [18,19,20,21], and photoelectrochemical (PEC) reactions [22,23,24,25] have been adopted to conduct the reduction of CO_2_, in which heat, light, or electricity were used to supply essential energy for the reaction. As is known, eight electrons are needed for each CO_2_ molecule to complete the conversion to hydrocarbon compounds. This leads to various products during the reduction process, resulting in complicated purification procedures and poor yield of desired products. Inspired by natural photosynthesis, highly efficient and specific enzymatic reactions were incorporated into the aforementioned reducing technologies to improve the efficiency and specificity of CO_2_ conversion [26,27]. 

In this review, research progress of CO_2_ conversion through different strategies were summarized and discussed with detailed comments of advantages and disadvantages (Scheme 1). In particular, using macrocycles as photosensitizers, current achievement, development, and catalytic activity in photocatalytic and photoelectrocatalytic reactions coupled with enzymatic reactions were highlighted. Finally, this article also presented the main challenges and certain future prospects for the conversion of CO_2_ to useful fuels.

## 2. Catalytic Reduction of CO_2_

### 2.1. Thermal Catalysis

Most thermal catalytic conversion of CO_2_ involves the hydrogenation reaction at relatively low temperatures (≤523 K) to produce useful fuels such as CO, methane, and methanol [28]. Since CO_2_ molecules are thermodynamically and chemically stable, large amounts of energy are required if CO_2_ is used as a single reactant. The introduction of other substances with higher Gibbs free energy (such as H_2_) as the co-reactant will make the thermodynamic process easier [29]. In the past few decades, great attention has been paid to the thermal catalysis of CO_2_ and significant progress has been made. Various catalysts, including metals (Cu, Co, etc.) and metal oxides (ZnO_2,_ InO_2_, etc.) [30,31,32] as well as novel nano-sized catalyst metal–organic frameworks (MOFs) have also been designed, prepared, and gradually developed. 

Rungtaweevoranit and coauthors [33] reported a catalyst where Cu nanocrystal (Cu NC) was encapsulated in a metal–organic framework UiO-66 [Zr_6_O_4_(OH)_4_(BDC)_6_, BDC = 1,4-benzenedicarboxylate] to form catalyst Cu⊂UiO-66, which could catalyze the generation of methanol via the hydrogenation of CO_2_. Figure 1a shows the synthetic process for an ordered UiO-66 crystal structure from the reaction of Zr oxide [Zr_6_O_4_(OH)_4_(−CO_2_)_12_] secondary building units (SBUs) and BDC. By using Zr(OPr^n^)_4_, one Cu nanocrystal was successfully encapsulated in a nanocrystal UiO-66, as shown in Figure 1b. Cu⊂UiO-66 was precisely placed on the Cu surface, yielding high interfacial contact between Cu NC and Zr oxide SBUs, which could ensure the reactants reach the active sites (Figure 1c). By comparing the methanol turnover frequency (TOF) of the Cu⊂UiO-66 catalyst with the Cu/ZnO/Al_2_O_3_ catalyst at different reaction temperature (Figure 1d), it was found that the performance of the catalyst exceeded that of the Cu/ZnO/Al_2_O_3_ catalyst at relative lower temperatures (<250 °C), which can steadily increase the conversion rate by 8 times and with 100% methanol selectivity.

Various carbon materials including carbon nanofibers [34], carbon nanotubes [35,36,37,38], biochar [39], and carbon felt [40] have also been employed as carriers for CO_2_ hydrogenation catalysts, taking advantage of their high hydrogen storage capacity, high thermal conductivity, and high specific surface area of carbon carriers. Nanosized materials are used to define nanoscale catalyst structures, in which the composition of catalysts and their surface structures can be adjusted and may bring to darewidespread applications. 

Despite such achievements, the reactant H_2_ added in the thermal catalysis process is usually more valuable than the product methane and methanol. Considering the higher cost than that from fossil fuels, the direct hydrogenation reaction of CO_2_ was rarely used to produce methane or methanol [28]. There are still great challenges in developing catalysts with high catalytic performance and long-term stability, reducing the size of thermal catalytic reactors and decreasing the production costs. In addition, more effective and economical methods to produce H_2_ are urgently needed [13]. In this case, CO_2_ conversion to useful fuels are attempted by other methods such as photocatalysis, electrocatalysis, and photoelectrocatalysis.

### 2.2. Photocatalysis

Solar energy is as an ideal energy source to replace traditional fossil fuels because it is an abundant, cheap, clean, and sustainable energy source. Therefore, the use of photocatalysts for solar-driven fuels or chemicals from CO_2_ is a very attractive approach. Similar to natural photosynthesis, electron–hole pairs are generated when the photocatalysts are exposed to solar light. The photogenerated electrons induce CO_2_ to undergo a redox reaction that results in hydrocarbon formation. There are three crucial procedures during the photocatalytic conversion of CO_2_: (1) absorption of sunlight; (2) charge separation and transfer; and (3) catalytic reduction of CO_2_ and oxidation of H_2_O [17]. Each procedure during the conversion of CO_2_ is closely related with the photocatalysts. Until now, the photocatalysts were mainly from semiconductor materials which are abundant on earth and easy to obtain [41]. As for the reaction products, CO, methane, formic acid, and other chemicals containing one or two carbon atoms are usually involved.

Until now, many efforts have been made to optimize the structure and composition of photocatalysts or integrate them with other functional units to construct multifunctional catalysts. For example, integration of photocatalysts with metal–organic frameworks (MOFs) has been demonstrated to offer more adsorptive sites for CO_2_ uptake because of their extreme larger surface area and microporous structure [42,43,44], resulting in remarkable improvement in CO_2_ conversion. However, there is a wide gap between the photocatalytic performance of these complexes and the requirements for practical application [45]. 

The construction of multi-junctions are randomly distributed on the surface of photocatalysts, which improve interfacial electron–hole separation and migration, even though the separation efficiency remains to be raised to a higher level [46]. Meng et al. [16] deposited MnO_x_ nanosheets and Pt nanoparticles on different facets of anatase TiO_2_ to form surface heterojunction. The results indicated that heterojunction with multiple nodes in the photocatalysts improved the conversion efficiency of CO_2_. 

Two dimensional (2D) nanosheets are particularly promising in improving charge separation because the photogenerated electrons and holes will move to the interface with shorter distances. Wei and coauthors [47] synthesized a series of heterostructured CdS/BiVO_4_ composites by depositing different amounts of CdS on the surface of BiVO_4_ nanosheets with variable thickness. The results showed that CdS/BiVO_4_ nanocomposites had higher photocatalytic activity in CO_2_ reduction than that of pure BiVO_4_ and CdS. Furthermore, the content of CdS in the composites were responsible for the yield of CO and CH_4_. Enhancement of photocatalytic activity was attributed to the synergistic effect of forming Z-scheme herterojunction and reduced thickness of BiVO_4_. According to density functional theory (DFT), theoretical calculations have been made for 2D photocatalysts and other types of catalysts [48,49,50,51,52], such that the characteristics of materials and the role of different components in the catalytic mechanism or the entire reaction cycle can be explored in depth. Computational approaches provide a way for understanding the catalytic effects at the mesoscopic and micro level. The mechanism of photocatalytic CO_2_ reduction remains obscure and needs more in-depth investigations.

In addition to inorganic photocatalysts, certain organic photocatalysts including porphyrin-based photocatalysts were also involved in CO_2_ reduction. Porphyrin are planar macrocyclic molecules and widely distributed in nature. There are four pyrroles connected in ring fashion through four methylene carbons. Also, porphyrins have highly delocalized π electrons to form a planar conjugated framework. This endows them with strong absorption in the visible light region and unique electronic redox characteristics. Moreover, the NH protons inside the ring are easily deprotonated and therefore exhibit remarkable coordination characteristics toward metal ions. For example, chlorophyll α is a complex of magnesium and porphyrin, and plays an important role in light capturing and H_2_O oxidation [53]. Incorporation of porphyrin into MOF can further improve its photocatalytic performance. Wang et al. [54] synthesized an indium–porphyrin MOF framework (In(H_2_TCPP)_(1-n)_[Fe(TCPP)(H_2_O)]_(1-n)_[DEA]_(1-n)_(In−Fe_n_TCPP-MOF) by incorporating porphyrin into MOF. Porphyrin rings in the MOFs support the single-site iron. This can both support the iron center as a catalytic active site and absorb visible light for high-performance conversion of CO_2_ to CO due to the synergetic effect between the porphyrin and the high-performance single-center Fe catalytic center (Figure 2). 

Due to strong chemical bonds of CO_2_ molecules and complex transformation path involving multiple electrons [55,56], many problems in the CO_2_ photocatalytic conversion need to be resolved, such as low conversion efficiency, poor selectivity of products, competition for the generation of H_2_, and rapid recombination of photogenerated electrons and holes [57,58]. Furthermore, most commercial photocatalysts used for CO_2_ conversion contain toxic and expensive transition metal elements, resulting in cost increase and waste disposal. Therefore, it is necessary to design and prepare new photocatalytic materials that can effectively increase the activity, suppress competitive reactions, improve conversion efficiency, and even develop a new CO_2_ catalytic reduction system. Researchers have shown that electric field can effectively inhibit charge recombination [59], so electrocatalytic reduction of CO_2_ has been extensively studied.

### 2.3. Electrocatalysis

Certain renewable resources such as solar energy and wind energy are usually intermittent and limited by the season and weather, so energy storage technology is necessary for uninterrupted energy supply [60]. The electrocatalytic conversion of CO_2_ to valuable chemicals is an attractive solution for reducing atmospheric CO_2_ and storing energy. Using an external electric field as an energy source and water as the proton donor, various catalysts are applied to catalyze the reduction of CO_2_. Compared with thermocatalysis, the electrocatalytic conversion is a higher cost-effective method because water replacing H_2_ is used as the proton donor. Electrocatalytic CO_2_ reduction has attracted great attention due to its mild operating conditions (normal temperature and pressure), controllable reaction process conditions and reaction rate, recyclable catalyst and electrolyte, high energy utilization, simple equipment, and achievable conversion efficiency [61,62,63,64]. In the past few years, researchers have explored electrocatalytic reduction of CO_2_ using different electrode materials, such as metals [65,66], transition metal oxides [19,67], transition metal chalcogenides [68,69], metal-free 2D materials [70,71], metal–organic frameworks (MOFs) [72,73,74], and various reduction products including CO, methane, formic acid, ethanol, and other compounds were obtained.

Hu et al. [75] investigated the electrocatalytic performance of cobalt meso-tetraphenylporphyrin (CoTPP) and its complex with carbon materials under both homogeneous and heterogeneous conditions. Their catalytic ability for CO_2_ reduction was significantly increased by the strong π–π interactions between CoTPP and carbon materials when CoTPP was incorporated with carbon nanotubes (CNTs) or similar carbon materials (Figure 3).

Wang and coauthors [76] designed and synthesized a series of stable reductive polyoxometalate-metalloporphyrin organic frameworks (M-PMOF, M = Co, Fe, Ni, zinc, as shown in Figure 4) by using reductive polyoxometalates (POMs, such as Zn-ε-Keggin cluster, as electron donor) as building block and metalloporphyrin as linker. Metalloporphyrin is helpful for electron mobility for its inherent macrocycle conjugated π-electron structure. Connection of Zn-ε-Keggin and M-TCPP might create an oriented electron transportation pathway by which multiple electron transfer processes in electrocatalytic CO_2_ reduction were completed. The electrocatalytic performance of M-PMOFs was measured by linear sweep voltammetry. The total current density of Co-PMOF at −1.1 V was 38.9 mA cm^−2^, higher than that of Fe-PMOF (25.1 mA cm^−2^), Ni-PMOF (20.02 mA cm^−2^), and Zn-PMOF (16 mA cm^−2^). These PMOFs, especially Co-PMOF, exhibited excellent electrocatalytic performance in CO_2_ reduction.

Davethu et al. [77] studied the electrochemical reduction of CO_2_ to CO on an iron–porphyrin center using a computational modeling. The results showed that a ligand, rather than metal reduction, resulted in stable binding of CO_2_ as an [Fe^III^ (CO_2_^2−^) (TPP^−^)]^2−^ complex during the reduction process. Subsequent proton transfer from phenol was considered as a proton-coupled electron-transfer process, and the second proton transfer does not change the electronic configuration of the metal complex. It was demonstrated that iron porphyrin was an effective catalyst and could efficiently transform CO_2_ to CO. In addition, the results indicated that CO_2_ binding was the rate-determining step in the reaction cycle, providing a promising direction for further optimization. A series of skeletons based on porous metal–porphyrin triazine were synthesized by trimming porphyrin units under ionothermal conditions [78]. The skeletons have high specific surface areas and homogenously dispersed transition metals. This facilitates the adsorption of CO_2_ molecules and exposure of larger number of active sites, thereby improving the performance of CO_2_ reduction. 

Reducing CO_2_ is, energetically, an uphill process. Many effective and selective homogeneous metal complex catalysts have been developed to promote CO_2_ conversion. For example, a number of pincer complexes can reduce CO_2_ into CO, CH_4_, or other compounds [79,80,81,82]. The ruthenium catalysts prepared by different methods are usually used in such systems [83,84,85,86] in which the turnover number (TON) of CO_2_ to methanol reached 9900 at an optimal condition. All of this shows the good conductivity and catalytic performance of pincer complexes, but these catalysts often rely on precious metals or supported ligands, such as bipyridines. Most catalysts used in CO_2_ reduction have the ability of hydrogen evolution, resulting in H_2_ generation during the conversion of CO_2_. This will suppress the formation of the desired products because of competition in electron capturing with the H_2_ evolution reaction. It is difficult to find a suitable electrocatalyst to selectively improve the conversion efficiency of CO_2_ [87]. Furthermore, multi-electron transfer process and electron transfer rate are the critical limiting factors in the electrocatalytic reaction. The main challenge is to find a highly selective electrocatalyst with high catalytic activity and long-term stability to overcome the thermodynamic stability of CO_2_ molecules. If electrocatalysis and photocatalysis are combined to take full advantage of their respective advantages, seem to be a better method to reduce CO_2_.

### 2.4. Photoelectrocatalysis

Since there is an inexhaustible solar energy supply in nature, it should be fully utilized in different ways. Photoelectrocatalysis, which combines the advantages of photocatalysis and electrocatalysis, is considered to be an ideal strategy for the selective conversion of CO_2_ into gaseous (such as CO, methane, etc.) and liquid products (such as formic acid, methanol, ethanol, etc.) under sunlight irradiation, and has therefore attracted great attention [88,89,90].

Photoelectrocatalysis makes the best use of solar energy to produce photoelectrons. The photogenerated electrons are transferred to the electrode surface under the action of an applied electric field, and finally obtained by CO_2_ for catalytic reduction. The applied electric field can effectively facilitate charge separation in the photocatalytic process [59], promote electron migration, and significantly improve the intrinsic activity and energy efficiency of CO_2_ molecules [91]. The efficient utilization of solar energy in photoelectrocatalysis can effectively overcome the problem of high energy consumption in the electrocatalysis of CO_2_.

In order to promote rapid charge transfer and improve the performance of photoelectrocatalysis, Ding and coauthors [92] patterned a photocathode through photolithography to expose a third of the surface, which is an effective and robust Si–Bi interface formed by Bi^3+^-assisted chemical etching of Si wafers and completed the reduction of CO_2_. This method increased the current density and facilitated the reduction of CO_2_ based on high product selectivity.

TiO_2_ is one of the most employed semiconductor in photo-assisted processes. Castro et al. [93] loaded different amounts of TiO_2_ on the photoanode using a Cu plate as the photocathode to build a photoelectric chemical device, and combined this with an electrochemical filter-press cell. This device was employed to continuously convert CO_2_ into alcohol with reducing energy consumption due to less external energy demand. Comparing the alcohol produced under different conditions, the TiO_2_ photoanode system exhibited enhanced alcohol production and reduced energy consumption under ultraviolet light irradiation. 

Different photocathodes have different light absorption capabilities, which essentially depend on the optical characteristics of the semiconductor. Table 1 lists and compares the performance of different photoelectrochemical systems of CO_2_ reduction from the latest literature.

The efficiency of CO_2_ conversion is a criterion of photoelectric conversion efficiency, which can be calculated by the following equation.

*Faradaic Efficiency (FE): FE* can be understood as the percentage of actual product/theoretical product.
(1)$$\;FE\left(\%\right)=\frac{e_{\mathrm{output}}}{e_{\mathrm{input}}}\times 100=\frac{n\left(\mathrm{mol}\right)\times m}{\frac{Q(\mathrm{Coulomb})}{F\left(\mathrm{Coulomb}/\mathrm{mol}\right)}}\times 100$$

In the above equation, n is the actual moles of product, m is the number of reaction electrons, Q is the calculated electric charge, and F is the Faraday constant (96,485 C/mol).

*Applied Bias Photon-to-Current Efficiency (ABPE): ABPE* is used to measure the efficiency when an external voltage ($V_{\mathrm{bias}}$) is applied.
(2)$$ABPE=\frac{J_{\mathrm{ph}}\left(\mathrm{mA}/{\mathrm{cm}}^{2}\right)\times \left[∆E^{{}^{\circ}}\left(V\right)-V_{\mathrm{bias}}\left(V\right)\right]\times FE}{P_{\mathrm{solar}}\left(\mathrm{mW}/{\mathrm{cm}}^{2}\right)}$$
where $J_{\mathrm{ph}}$ is the photocurrent detected under the external voltage, $∆E^{{}^{\circ}}$ is the thermodynamic energy stored in the PEC reactor, and $P_{\mathrm{solar}}$ is the power density of light.

### 2.5. Enzyme

As is known, the catalyst is one of the key components in CO_2_ reduction systems including thermal catalysis, electrocatalysis, photocatalysis, and photoelectrocatalysis. However, a prominent problem associated with most catalytic systems is low product selectivity, where more than one product, including CO, formate, methane, ethylene, and other components, are usually observed in one catalysis reaction. By contrast, the reduction of CO_2_ via biocatalytic processes received particular attention because of their special substrate and product selectivity as well as high conversion efficiency.

Enzymes are biocatalysts renowned for their high efficiency and selectivity. In living cells, different enzymes often work together or in a specific order to catalyze multi-step biochemical reactions, playing crucial roles in the synthesis of natural products and metabolism [103]. Inspired by the biocatalytic reaction, enzymes including enzyme cascades were explored in vitro to complete the conversion of CO_2_ to certain chemicals via a one-step or multi-step process. Figure 5 shows the approximate number of papers published in the past two decades using enzymes as catalysts in CO_2_ conversion. It is obvious that the research has presented an increasing tendency, especially in the recent 10 years, suggesting more and more attention was paid to the biocatalytic conversion of CO_2_.

In 1993 and 1994, Yoneyam et al. [104,105] demonstrated that CO_2_ can be biocatalyzed into CH_3_OH in a CO_2_-saturated phosphate buffer solution, in which pyrroloquinoline quinone (or methyl viologen) was used as an electron mediator, and formate dehydrogenase, formaldehyde dehydrogenase, and alcohol dehydrogenase were used as biocatalysts. Subsequently, Obert [106] presented the reduction of CO_2_ to methanol using three different dehydrogenases in three consequent reductions, in which reduced nicotinamide adenine dinucleotide (NADH) molecules were required at each step. Such a multi-enzyme system was composed of three different dehydrogenases (Figure 6) that catalyze the conversion of CO_2_ to CH_3_OH in the presence of NADH. In this enzyme cascade, formate dehydrogenase (FDH) catalyzes the conversion of CO_2_ to formate, formaldehyde dehydrogenase (FaldDH) then catalyzes the formate to formaldehyde and, finally, alcohol dehydrogenase (ADH) catalyzes the formaldehyde to CH_3_OH. Each enzymatic step in the reduction cascade proceeds in the opposite direction of the natural (reversible) enzyme-catalyzed reaction and requires NADH as the electron donor for the reaction.

Researchers [107] also compared CO_2_ reduction from different sources of FDH, F_ald_DH, and ADH to gain an in-depth understanding of the multi-enzyme cascade reaction. The formate dehydrogenase (ClFDH), formaldehyde dehydrogenase (BmFaldDH), and alcohol dehydrogenase (YADH) were from *Clostridium ljungdahlii*, *Burkholderia multivorans*, and *Saccharomyces cerevisiae*, respectively. A 500-fold increase in total turnover number was observed for the ClFDH–BmFaldDH–YADH cascade system compared to the *Candida boidinii* FDH–*Pseudomonas putida* FaldDH–YADH system. This is conducive to develop an enzyme cascade reaction with higher conversion efficiency. The three dehydrogenases can not only be combined to convert CO_2_ into methanol but also can be used individually to convert CO_2_ to corresponding products such as formate or formaldehyde. 

Up to now, dehydrogenases including formate (FDH), formaldehyde (FaldDH), and alcohol dehydrogenase (ADH) that are usually used as biocatalysts in CO_2_ reduction are NAH(P)H dependent. However, NADH is expensive, and the extensive use of NADH increased the cost of enzymatic reaction. The regeneration efficiency of NAD(P)H became an important criterion for evaluating the biocatalytic reaction and more efforts should be devoted to improving the NAD(P)H yield and reducing the production cost. 

The combination of enzymes and photocatalysts for CO_2_ conversion has attracted increasing attention because it makes full use of the abundant energy supply of solar light and high specificity of enzyme catalysis [108,109,110]. These reactions can be conducted at mild conditions similar to the photosynthesis that occurs in plants or certain bacteria. This is also called artificial photosynthesis. The combination of electrodes with suitable biological enzymes could minimize the requirement of high overpotentials to excite the electrocatalytic reaction of CO_2_. Such an electroenzymatic reaction can be carried out at a low overpotential, or even without external bias [111,112,113].

### 2.6. Enzyme Coupled to Photocatalysis

The photosynthesis that occurs in green plants and certain bacteria converts solar energy into chemical energy that can be well utilized by organisms and, at the same time, absorbs carbon dioxide and produces oxygen to maintain the carbon–oxygen cycle on the earth. This inspired people to explore the intrinsic mechanism of the photosynthesis process and to construct artificial analogues via biomimetic mythologies for alternative sustainable energy carriers instead of traditional fossil fuels. Conversion of solar energy to chemical energy consists of hydrogen production, oxygen evolution, and carbon dioxide reduction as well as nitrogen fixation. Among them, the conversion of the human-made greenhouse gas CO_2_ into valuable fuels/chemicals using solar energy is considered a promising and compelling approach to solar energy utilization because it aims to simultaneously solve problems regarding global energy and environment [114]. In particular, photoreaction coupled with enzymes provides a highly efficient, specific, and energy saving strategy for CO_2_ conversion and has attracted special attention in recent years [115]. 

Yadav and coauthors [116] developed a graphene-based visible light active photocatalyst–FDH coupled system in which CO_2_ was specifically converted to formic acid. The chromophore (multianthraquinone-substituted porphyrin, MAQSP) with chemically modified graphene (CCG) were covalently combined to form a new catalyst (CCGMAQSP), through which the light-harvesting efficiency can be enhanced. The chromophore absorbs sunlight and acts as an electron donor. The light-generated electrons are transferred to the organometallic rhodium complex through graphene (electron acceptor). The rhodium complex accepts electrons from graphene and thus is reduced and further extracts H^+^ from water. NAD^+^ accepts electrons and H^+^ to form NADH [89]. Formate dehydrogenase converts CO_2_ to formate in the presence of NADH, as shown in Figure 7.

In addition to rhodium complexes, methyl viologen (MV) was also used as an electron mediator. Kumar and coauthors [117] designed a photocatalyst of graphene oxide modified with cobalt metallized aminoporphyrin (GO-Co-ATPP) for conversion of CO_2_ to formic acid under visible light. Porphyrin captures photons and generates electrons and transfers to methyl viologen (MV) complex via graphene. The organometallic MV complex can easily obtain electrons and exist in its reduced form. It further extracts protons from the aqueous solution, and transfers electrons and hydrogen ions to NAD^+^, which finally transforms to NADH, which is used for CO_2_ reduction.

However, this system complete the photocatalytic reaction and enzymatic reaction in the same environment, which will have a certain impact on the stability and activity of enzyme. For example, FDH has attracted much attention in recent years because it can directly reduce CO_2_ to formate without any other byproducts. FDH is divided into two types according to cofactor requirements: NADH-dependent FDH and metal-dependent FDH. Although metal-containing FDHs have a higher catalytic activity for CO_2_ reduction, these NADH-independent FDHs contain extremely unstable oxygen components, such as metal ions (tungsten or molybdenum), iron–sulfur clusters, and selenocysteine. The oxidation reaction of water may occur during photocatalysis to generate oxygen, which will affect the enzyme catalytic activity in the system, affect the stability of FDH, and then affect the final product of formic acid. The low compatibility of the photocatalysis and the biocatalysis in the system hindered its development.

As is known, thylakoids in chloroplast were employed to couple the photoreaction and the biological reaction system by which the enzymatic reaction was separated from the water oxidation reaction to protect enzymes from inactivation. In order to achieve cooperation between photocatalysis and biocatalysis and improve compatibility, Zhan et al. [118] developed an artificial thylakoid by decorating the inner wall of protamine–titania (PTi) microcapsules with cadmium sulfide quantum dots (CdS QDs), and coupled with biocatalysis to form an artificial photosynthesis system. Cds QDs absorb visible light and generate electrons and holes. The electrons are transferred to the outer surface of the capsule through the heterostructure of Cds QDs and amorphous TiO_2_. Through the intermediate transfer of the rhodium complex, formate dehydrogenase converts CO_2_ to formic acid in the presence of NADH. The size-selective capsule wall separates photocatalytic oxidation and enzymatic reduction of CO_2_, thereby protecting the enzyme from inactivation that usually caused by photogenerated holes and active oxygen.

Most enzymes are powdered reagents, which makes them difficult to separate from the substrate and cannot be recycled. It can effectively reduce the costs and simplify the product purification process through enzyme immobilization. Zeolite imidazolate framework (ZIF) is a type of MOF material that possesses well-defined pore structure, excellent chemical–thermal stability, extremely high surface area, and other excellent properties [119]. Moreover, ZIF is easy to prepare and has little effect on enzyme activity because the preparation is usually conducted at mild conditions. It has become one of the common methods for enzyme immobilization [120]. Zhou et al. [121] combined ZIF and TCPP to construct a photocatalytic multi-enzyme cascade biomimetic carbon sequestration system (TCPP&FF@ZIF-8 (FF = FateDH and FaldDH)). TCPP was used as the photocatalyst and ZIF-8 as the multi-enzyme immobilized carrier for FateDH and FaldDH. The catalytic system was then used to absorb CO_2_ and transform it to chemicals. Interestingly, the repeated stability investigation of the composite system showed that the residual activity of 3% TCPP and FF@ZIF-8 remained at 52.93 % after 10 batches of repeated use, suggesting that the system had excellent structural stability, light stability, and cycling stability.

Using different photocatalysts, enzymes, and cofactors, various products and yields were obtained. Table 2 provides a simple comparison of the performance of different coupled photocatalytic/enzymatic CO_2_ reduction systems in recent years. 

The combination of photocatalysis and biocatalysis showed higher efficiency in CO_2_ reduction than that of photocatalysis. However, the prominent problem is the interference between the photocatalysis and biocatalysis, resulting in corrosion of the photocatalyst and inactivation of the enzyme. Additionally, the enzyme used in the biocatalysis is reversible and it is easy to perform the reverse reaction and charge reorganization [128]. To solve this problem, researchers adopted an electric field on the basis of photoenzymatic catalysis, which can effectively promote charge separation and improve the conversion efficiency of CO_2_.

### 2.7. Enzyme Coupled to Photoelectrocatalysis

Photocatalysis and electrocatalysis are combined to form a photoelectric cooperative catalysis, and then combined with biocatalysis to form a photoelectrochemical (PEC) cell similar to maintain the optimal conditions of enzymes and improve conversion efficiency [129,130]. Conducting wire was used to ensure the oriented transfer of reducing equivalents (primarily electrons, H^+^) from the photoelectrode to biocatalysis. Nam et al. [131] imagined that photoelectrochemical (PEC) cells could inhibit biocatalytic charge recombination and reverse reactions because photocatalytic and biocatalytic reactions can be separated in the anode and cathode compartments, respectively. An anode compartment with cobalt phosphate (Co-Pi) deposited hematite (Fe_2_O_3_) photoanode for photocatalytic water oxidation and a cathode compartment with formate dehydrogenase for NADH regeneration and CO_2_ reduction were designed (Figure 8). The co-catalyst (Co-Pi) in the photoelectrode can reduce the activation energy quickly, improve the quantum efficiency by promoting charge separation, and consume the photogenerated charges in time to improve the stability of the photoelectrode [132]. As can be seen from Figure 8, the PEC system is divided into two compartments in which the oxidation reaction of water can be well separated from the enzymatic reaction. The two compartments do not affect each other, enabling the formate dehydrogenase to work at its optimal pH conditions.

Generally, enzymes are easily affected by the reaction environment, making the enzyme electrode unstable [133]. For example, the enzyme cannot perform its maximum activity at a non-optimal pH solution. Eun-Gyu Choi and coauthors [134] studied the effect of pH on a coupled photoelectric–enzyme system. RcFDH-driven CO_2_ reduction was predominant at an acidic pH, whereas formate oxidation was favorable at basic conditions. pH = 6.5 is the most suitable condition for CO_2_ reduction to formic acid in their photoenzyme system by the comparison of the results at different pH values. For the stability and reusability of the enzyme, appropriate fixation methods can be adopted. Lee et al. [112] reported that a tightly organized biophotocathode (EC-PDA)-electrochemically synthesized polydopamine (PDA) film was copolymerized with FDH (E) and NADH (C) in which CO_2_ can be reduced to formic acid with high selectivity. The PDA was chosen as the substrate for enzyme immobilization because of its excellent biocompatibility and charge transfer ability [135]. The PDA layer on the electrode can fulfill the requirement of electron transfer and enzyme stabilization and extend the service life of the enzyme [136]. A similar photoelectrochemical cell was constructed by the EC-incorporated PDA bioelectrode and cobalt phosphate/bismuth vanadate (CoPi/BiVO_4_) photoanode by which the reduction of CO_2_ can be achieved without external bias. 

In another study, a voltage was applied to the constructed PEC cell-Co-Pi/Fe_2_O_3_ photoanode and BiFeO_3_ photocathode [137]. The polarization treatment drove surface charge accumulation and accelerated the transfer of electrons to the electrolyte, therefore resulting in an improvement in CO_2_ conversion efficiency. The tandem PEC cell with an integrated enzyme cascade (TPIEC) system mimics the natural photosynthetic Z-scheme for the biocatalytic reduction of CO_2_ to methanol. The rate of methanol production per unit of reaction volume and the rate of methanol production per unit mass of photocatalyst can reach 220 μM h^−1^ and 220 μM g_cat_^−1^ h^−1^, respectively. This device exhibited significantly higher rate of methanol than those of other studies. Due to the relatively high price of the metal rhodium complex [138], researchers further improved the structure of the photoelectrochemical cell to reduce the cost. Neutral red was used as an alternative electron mediator to replace the metal rhodium complex and was conducive to electron recycling between the electrode and NAD^+^ [139]. Sokol and coauthors [140] adopted a semi-artificial design. The cathode containing formate dehydrogenase was connected to the photoanode containing photosynthetic water oxidase (photosystem II) to achieve the metabolic pathway of formic acid that is formed by light fixation of CO_2_ in the absence of precious metal catalyst (Figure 9). This demonstrated the feasibility of the nonmetal catalysts for the conversion of CO_2_ to formic acid and provides a novel method for CO_2_ photoelectrochemical reduction. Table 3 lists the performance parameters of different photoelectrochemical/enzyme systems.

Researchers not only improved the noble metal mediator, but also further studied the coenzyme NADH. NADH is necessary in most biocatalytic reductions of CO_2_ to provide electrons for dehydrogenase. NADH is expensive and easily forms enzymatically inactive dimers, NAD_2_, resulting in reduction of enzyme activity [143]. It is therefore necessary to develop economical methods for NADH cofactor regeneration. To achieve this goal, chemical, photochemical, and electrochemical regeneration of NADH has been developed over the last few decades [122,144,145,146]. Electrochemically mediated electron injection into the enzyme not only bypasses the requirement of NADH but also simplifies product separation [147]. Amao and Shuto [148] have demonstrated the use of viologen-modified FDH immobilized on an ITO electrode to electrochemically convert CO_2_ to formic acid. This study showed evidence of electrons directly transferring to FDH active sites from electrodes and represents a new strategy for CO_2_ reduction in the absence of NADH. However, finding alternative compounds to replace NADH is also an attractive research direction. Hence, researchers have tried to use cheaper electron mediators, such as methyl viologen (MV^2+^), instead of NADH. Miyatani et al. [149] developed a system using MV which combined the synthesis of formic acid from CO_2_ (bicarbonate ion) with FDH and MV^2+^, and photoreduction with ZnTMPyP as photosensor and TEOA as electron donor. Formic acid was successfully formed from bicarbonate ions and FDH in the absence of NADH. 

Moreover, investigations have demonstrated that the oxidation of formate to carbon dioxide does not occur readily with MV^2+^. Therefore, the overall yield of formate would be preserved without loss from reoxidation [150]. Ishibashi et al. [151] used methyl viologen (MV^2+^) instead of NADH. A photoelectrochemical system was composed of TiO_2_ nanoparticles as photocatalyst, MV^2+^ as electron carrier, and FDH as biocatalyst, in which CO_2_ was successfully reduced to formic acid without sacrificing reagent, external bias, and NADH. 

## 3. Conclusions and Outline

Thermal catalysis, photocatalysis, electrocatalysis, photoelectrocatalysis, and enzyme catalysis can effectively alleviate the greenhouse gas CO_2_. The catalyst is the key component of different CO_2_ reducing systems. A suitable catalyst can not only reduce energy consumption but also facilitate generation and transfer of electrons. On one side, from the perspective of energy consumption, we wish to complete the conversion of CO_2_ while consuming as little energy source as possible to save energy and the environment. Therefore, photocatalysis and photoelectrocatalysis that adopt clean and sustainable solar energy to drive the conversion are of interest, in which porphyrin-based macrocycles or their combination with other components presented promising properties of light-capturing and charge transferring. On the other side, the selectivity of products has challenged the development of CO_2_ conversion. Specific product generation from CO_2_ will greatly reduce the cost of subsequent product separation and purification. This has inspired researchers to pay particular attention to biocatalysts because of their high specificity and efficiency in catalyzing biochemical reactions. Among the aforementioned methods, enzyme coupled to photocatalysis and enzyme coupled to photoelectrocatalysis has integrated the two sides successfully, showing great potential in solar energy utilization and specific conversion of CO_2_. They are worthy of more investigation to make biocatalysis compatible with photocatalysis or photoelectrocatalysis. In general, the conversion of CO_2_ to valuable fuels or chemicals appears to have a bright future, and continuous efforts are needed to improve the catalytic efficiency, conversion rate, and product selectivity.

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
