# Peer review of "Research Progress in Conversion of CO2 to Valuable Fuels"

_molecules, 2020, doi:10.3390/molecules25163653_

Round 1
Reviewer 1 Report
This manuscript is of great interest and originality in the hot field of CO2 conversion to solve an environmental problem and generate products with high added value. Although a large number of reviews have been published on this topic, since 2006-2020, mainly focused on catalytic, electrocatalytic, and photo-electrocatalytic processes, the authors have opted for reviewing biomimetic processes by using macrocycle materials. Indeed, a bird's-eye review of the use of enzymes in CO2 conversion reports around 200 published works. Therefore, I consider that this work could be accepted for its publication, taking into account the following recommendations:
(1) Attach and discuss a bibliometric study showing the state-of-the-art on the use of enzymes in CO2 conversion.
(2) Attach and discuss the different tables that clearly illustrate the results of CO2 reduction as follows:
Table 1. Coupled photocatalytic/enzymatic systems...;
Table 2. Coupled electrocatalytic/enzymatic systems... and
Table 3. Coupled photo-electrocatalytic/enzymatic systems...
(3) Analyze and discuss in more depth the aspects of selectivity and stability in the sections: 2.6 and 2.7.
(4) Authors use terms like, "photoenzyme coupling catalysis and photo-electroenzyme catalysis", I think it would be more appropriate to use, "enzyme coupled to photocatalysis and enzyme coupled to photo-electrocatalysis", for instance, buy anyway, is my point of view, which could be debatable.
Reviewer 2 Report
The paper presents a review of the catalysts for use in the the CO2 reduction. The paper is well written and well organized, however it seems that it is missing many important papers that have been published in the field.
1/ My main objection is the complete omission of the homogeneous catalytic systems such as the ones based on the pincer complexes. The porphyrins that are discussed, which belong to the molecular catalysts are only presented in immobilized setups.
2/ As a theoretician, i would be happy to see the review of the computational approaches to this process.
3/ Suprisingly little discussion is devoted to the comparison of different systems and approaches.
